# Impact of *Campylobacter* spp. on the Integrity of the Porcine Gut

**DOI:** 10.3390/ani11092742

**Published:** 2021-09-19

**Authors:** Alexandra Rath, Silke Rautenschlein, Janina Rzeznitzeck, Gerhard Breves, Marion Hewicker-Trautwein, Karl-Heinz Waldmann, Alexandra von Altrock

**Affiliations:** 1Clinic for Swine and Small Ruminants, Forensic Medicine and Ambulatory Service, University of Veterinary Medicine Hannover, Foundation, 30173 Hannover, Germany; Karl-Heinz.Waldmann@tiho-hannover.de (K.-H.W.); Alexandra.von.Altrock@tiho-hannover.de (A.v.A.); 2Clinic for Poultry, University of Veterinary Medicine Hannover, Foundation, 30559 Hannover, Germany; Silke.Rautenschlein@tiho-hannover.de (S.R.); Janina.Rzeznitzeck@tiho-hannover.de (J.R.); 3Institute for Physiology and Cell Biology, University of Veterinary Medicine Hannover, Foundation, 30173 Hannover, Germany; Gerhard.Breves.iR@tiho-hannover.de; 4Department of Pathology, University of Veterinary Medicine Hannover, Foundation, 30559 Hannover, Germany; Marion.Hewicker-Trautwein@tiho-hannover.de

**Keywords:** pig, intestine, *Campylobacter*, experimental infection, Ussing chamber

## Abstract

**Simple Summary:**

Campylobacteriosis is recognised as a leading food-borne zoonotic enteric disease of humans, mainly caused by *Campylobacter jejuni*, and to a minor extent by *C. coli*. In general, pigs are symptomless carriers primarily of *C. coli*, but may also harbour *C. jejuni*. In a swine infection model, weaned piglets were intragastrically inoculated with *C. coli* (ST-5777; *n* = 8), with *C. jejuni* (ST-122; *n* = 10), or with both strains (*n* = 8) and 11 piglets served as control. The health status was monitored and the influence on the intestinal barrier was investigated using the Ussing chamber technique and histological examinations. After inoculation, no clinical manifestations were noted. No gross lesions were observed during dissection four weeks post inoculation, and no pathohistological changes were detected in the intestinal mucosal sections. On the other hand, in the caecum of *C. jejuni* mono-inoculated pigs, we recognized an impact on transepithelial transport processes. We observed an increased Cl^−^ secretion by using the Ussing chamber technique.

**Abstract:**

*Campylobacter* (*C.*) is the most common food-borne zoonosis in humans, which mainly manifests with watery to bloody diarrhoea. While *C. jejuni* is responsible for most cases of infection, *C. coli* is less frequently encountered. The object of the study was to prove the clinical impact of mono- and co-colonisation of *C. coli* and *C. jejuni* on weaned piglets in an infection model and to investigate the impact on transepithelial transport processes in the jejunum and caecum. At an age of eight weeks, eight pigs were infected with *C. coli* (ST-5777), 10 pigs with *C. jejuni* (ST-122), eight pigs with both strains, and 11 piglets served as control. During the four-week observation period, no clinical signs were observed. During dissection, both strains could be isolated from the jejunum and the caecum, but no alteration of the tissue could be determined histopathologically. Mono-infection with *C. jejuni* showed an impact on transepithelial ion transport processes of the caecum. An increase in the short circuit current (I_sc_) was observed in the Ussing chamber resulting from carbachol- and forskolin-mediated Cl^−^ secretion. Therefore, we speculate that caecal colonisation of *C. jejuni* might affect the transport mechanisms of the intestinal mucosa without detectable inflammatory reaction.

## 1. Introduction

*Campylobacter (C.)* has replaced *Salmonella* as the most common human diarroheal pathogen in all developing and non-developing countries [1,2]. In 2019, 61,526 cases were registered in Germany [3]. While it is part of the intestinal flora without clinical influence [4] in a large number of mammals and birds, the bacterium frequently triggers acute enteritis in humans [1,2]. This enteritis is primarily associated with *Campylobacter jejuni* (73%) and mainly transmitted by poultry meat, followed by *C. coli* (10%) with pork as a potential vector [3]. The prevalence of *Campylobacter* in pig herds is estimated to be as high as 100% [4,5], where *C. coli* distinctly predominates over *C. jejuni* [6]. In most cases, the infection is transmitted faecal-orally from sow to piglet [7].

In general, *Campylobacter* is considered a commensal of the intestinal microbiota in pigs, but gnotobiotic- and colostrum-deprived young piglets developed clinically manifested enteritis after inoculation with *C. jejuni* [8,9,10,11]. Immune reactions and lesions have also been described in broilers, hens, and turkeys after infection [12,13]. Host factors, especially the composition of the mucus layer, as well as microbial characteristics, such as the specific nutrient metabolism, seem to influence the outcome of an infection [14,15,16].

The pathogenicity of the bacterium is mainly due to its motility, chemotactic orientation, adhesion and invasion ability, and toxin production [17].

In cell cultures (T84, Caco-2 cells), increased permeability of the intestinal mucosa, decreased transepithelial resistance, and alteration of tight junctions were observed after infection with *Campylobacter* spp. [18,19]. In addition, studies of the barrier properties of the chicken intestine using the Ussing chamber revealed changes in the intestinal barrier despite the absence of symptoms. After adding histamine, the infected group showed decreased ion transport, indicating decreased nutrient uptake. Furthermore, the infection caused intestinal histomorphological changes in the jejunum, characterised primarily by shortened villi and decreased crypt depth [20,21]. In this study, glucose, forskolin, and carbachol were used to investigate transport mechanisms in the Ussing chamber. The addition of glucose is used to examine sodium-glucose-linked transporter 1 (SGLT1) in jejunal tissue [22,23]. Forskolin activates cAMP-driven Cl^−^ secretion [24], while carbachol (used only in the caecum) acts on Ca-driven secretion [25,26,27]. Histological examinations were performed after haematoxylin and eosin (H&E) staining.

Comparable investigations on the gut integrity and function of the gut epithelium of swine after *Campylobacter* colonisation do not yet exist for the pig and were therefore performed in the present study.

## 2. Materials and Methods

### 2.1. Campylobacter Strains and Inoculum Preparation

Two different *Campylobacter* strains were used for this experiment: *C. coli*: ST5777/CT828 and *C. jejuni*: ST122/CT206. Both strains were isolated from poultry and had been used in a previous study [28]. The genotypes, characterised by multilocus sequence typing (MLST), had previously been isolated several times in association with human disease (http://pubmlst.org/campylobacter/, accessed on 18 May 2021). The *C. coli* strain was isolated from a sporadic case of gastroenteritis in humans in Luxembourg in 2012. The *C. jejuni* strain was isolated from faeces of a human case associated with Guillain-Barré syndrome. It has also been found in numerous cases in the United States, the Netherlands, and Germany in association with gastroenteritis in humans. Most recently, the strain was isolated in one case of systemic disease in the United Kingdom.

Different laboratory-induced antibiotic resistance patterns of the strains enabled differentiation when grown on agar supplemented with the corresponding antibiotic substance. The *C. coli* ST-5777 strain is resistant against nalidixic acid, and *C. jejuni* ST-122 against streptomycin.

For the infection experiment, nutrient broth (nutrient broth no. 2, CM0067, OXOID/Thermo Scientific Inc., Waltham, MA, USA) was enriched with bacterial colonies of the respective *Campylobacter* spp. 48 h before infection and incubated in a micro-aerophilic atmosphere (5% O_2_, 10% CO_2_ and 85% N_2_) with Thermo Scientific™ Oxoid™ CampyGen™ (Thermo Scientific Inc.) at 37.4 °C. An infective dose of 10^8^ colony-forming units (cfu) per animal in 10 mL broth was targeted. The actual infectious dose was determined subsequently by a dilution series.

### 2.2. Animals and Experimental Set-Up

In total, 37 (21 male, 16 female) crossbred piglets ((Danish Landrace × German Large White) × German Landrace) were used for the experiment. They were derived by Caesarean section to exclude a *Campylobacter* field colonisation and raised in isolation units. The pigs were housed and treated in accordance with the German Animal Welfare Act which complies with the German Research Council’s criteria and the EC Directive 2010/63/EU for animal experiments. The absence of *Campylobacter* spp. was ensured by regular rectal swabs. Strict hygiene standards were established to avoid transmission of *Campylobacter* between units after infection.

CulinaMilk^®^ (H. Bröring GmbH & Co. KG, Dinklage, Germany) was used as milk replacer. After weaning in the fifth week of life, the pigs were fed twice daily with a conventional rearing diet (Deuka GmbH & Co. KG, Düsseldorf, Germany) and had access to drinking water *ad libitum*. The piglets were injected with 2 mL iron dextrane (Ursoferran^®^ 100 mg/mL, Serumwerk Bernburg AG, Bernburg, Germany) subcutaneously (s.c.) on their third day of life.

The piglets were randomly divided into four different groups (Table 1) and inoculated with the respective *Campylobacter* strain in week eight. Hereafter, the control group is referred to as Group 0, the group infected with *C. coli* is called Group 1, and the group infected with *C. jejuni* is called Group 2. In addition, the group infected with both strains is called Group 3.

In the eighth week of life, the animals were anaesthetized with azaperone (2 mg/kg body weight Stresnil^®^, Elanco^TM^, Elanco Inc., Greenfield, IN, USA) and ketamine (20 mg/kg body weight Ketamin^®^, CP-Pharma^®^, CP-Pharma Handelsgesellschaft GmbH, Burgdorf, Germany) and inoculated by intragastric application via a stomach tube (CH 12, B. Braun Melsungen AG, Melsungen, Germany), according to the scheme in Table 1.

During the experiment, the pigs underwent daily monitoring including evaluation of the general health condition and rectal measurement of body temperature in the morning. If the rectal temperature rose above 40 °C, the temperature was rechecked after 12 h. If it was still elevated thereafter, the piglets were treated orally with Metacam^®^ (Meloxicam, 15 mg/mL; 0.4 mg/kg body weight, Boehringer Ingelheim Vetmedica GmbH, Ingelheim, Germany).

At necropsy four weeks after inoculation, the animals had an average body weight of 30.9 ± 5.6 kg.

### 2.3. Sampling and Cultivation of Campylobacter Strains from Faeces and Ingesta

During the experimental period, rectal swabs were collected both before and after infection to demonstrate the absence of *Campylobacter* spp. and to confirm intestinal colonisation after infection, respectively.

All animals were stunned by captive bolt shot and killed by exsanguination in the 13th week of life (four weeks post-inoculation with *Campylobacter* spp.). Immediately thereafter, the abdomen was opened by a midline incision and the intestinal tract was removed. During dissection, the intestinal tract was examined macroscopically for pathological abnormalities.

The first three metres of jejunum were discarded, and the following 30 cm were removed. Additionally, the middle part of the caecum was excised for Ussing chamber experiments.

Previous experiments showed that *Campylobacter* spp. colonise mainly in these two intestinal sections [28]. Therefore, the experiments were carried out on the jejunum and caecum using the Ussing chamber.

The intestinal segments were rinsed with cold physiological saline solution (4 °C, 0.9% NaCl (*w*/*v*)) to remove ingesta and afterwards stored in cooled serosal buffer solution (Table 2) for Ussing chamber experiments. After longitudinal incision of the jejunum along the mesenteric border, the mucosa was stripped of the underlying muscle layers. The tissue from the caecum was also opened and the mucosa separated. The specimens were then clamped in the Ussing chambers.

For the histological examinations, samples were taken from the duodenum, jejunum, ileum, caecum and colon. The sampling sites of the jejunum and caecum were adjacent to the samples taken for the Ussing chamber. Intestinal samples were rinsed with physiological saline solution and placed in 4% (*w*/*v*) phosphate buffered formalin for at least 48 h.

For the microbiological examination, faecal and intestinal samples (intestinal mucosa + intestinal contents) were placed in a tube filled with 10 mL Preston broth for enrichment and incubated for 48 h at 37.4 °C in a micro-aerophilic atmosphere (5% O_2_, 10% CO_2_ and 85% N_2_). After this time, the enrichment was spread onto modified charcoal cefoperazonedeoxycholate agar plates (mCCDA, Thermo Scientific Inc., Waltham, MA, USA) and incubated again for 48 h as described above (the procedure can be seen in Appendix A). The CCDA agar was enriched with the corresponding antibiotic (*C. coli* = nalidixic acid; *C. jejuni* = streptomycin; detailed ingredients can be found in Appendix A). Subsequently, the presence of *Campylobacter*-specific colonies was examined, and a native smear was performed if the results were questionable. In addition, random samples from Groups 1 and 2 were spread on the CCDA agar enriched with the other antibiotic in order to exclude cross-contamination of the mono-infection groups.

### 2.4. Set-Up for Ussing Chamber Technique

Stripped jejunal and caecal mucosal tissues were mounted in the Ussing chambers. Three chambers were used per intestinal segment and per animal (exposed serosal area of 1.00 cm^2^). By mounting the tissues between the two halves of the Ussing chamber, a barrier was formed between the serosal and mucosal compartment, each of which were filled with 10 mL buffer. For the serosal side, the same buffer was used for the jejunum and caecum. On the mucosal side, different buffers were used for the two intestinal segments. The buffer composition is shown in Table 2.

All buffer solutions had an osmolarity of 297 mosmol kg^−1^, were heated to 37 °C, and were continuously aerated with carbogen. The pH value ranged between 7.45 and 7.47.

All chemicals for the buffer solutions were obtained from Merck KGaA, Darmstadt, Germany, except mannitol, which was purchased from Sigma-Aldrich Inc., St. Louis, MO, USA, and HEPES, which was purchased from Carl Roth GmbH + Co.KG, Karlsruhe, Germany.

After filling the chambers with buffer solutions, they were connected to computer-controlled voltage clamps (K. Mussler Scientific Instruments, Aachen, Germany) through which the transepithelial potential difference (PD_t_), tissue conductance (G_t_), and short circuit current (I_sc_) were measured. The experiment was performed under short-circuited conditions.

Firstly, the tissues were equilibrated for 30 min after attaching the voltage clamps and filling in the buffers. After the 30-min equilibration phase, the voltage was clamped to zero. The further experiment was performed in accordance with two different protocols, respectively for the tissue from the jejunum and caecum.

According to the first protocol, the mucosal glucose concentration of the jejunal tissues was adjusted to 10 mmol/L after equilibration. In order to avoid transepithelial osmotic gradients, this was compensated by adding mannitol to the serosal side. After an incubation period of 30 min, 5 × 10^−6^ M forskolin (1 µL, serosal) was added.

The second protocol was applied to the caecal tissues: 10^5^ M carbachol (10 µL, serosal) was added, incubated for another 30 min and then 5 × 10^−6^ M forskolin (1 µL, serosal) was added. Forskolin and carbachol were obtained from Sigma-Aldrich Inc., St. Louis, MO, USA. After adding forskolin, another 30 min incubation period was performed before the experiment was terminated.

### 2.5. Histological Investigation

Tissue samples from duodenum, jejunum, ileum, caecum, and colon were stored in formalin solution (100 mL formaldehyde 37%, 900 mL bidest, 4 g NaH_2_PO_4_H_2_O, 6.5 g Na_2_HPO_4_) for at least 48 h. Post-fixation, tissue drainage and embedding of the fixed tissue with paraffin was performed in the Shandon Pathcentre^®^ Tissue Processor (Thermo Scientific Inc., Waltham, MA, USA). The samples were then poured into paraffin wax blocks at a paraffin pouring station (TES 4004 Tissue Embedding Station, PFM, Cologne, Germany). Tissue sections with a thickness of 2 μm were prepared from the paraffin wax blocks on a slide microtome (2646, Reichert & Jung GmbH, Heidelberg, Germany). The sections were placed in a water bath at 40 °C, applied to Menzel Superfrost^®^ Ultra Plus slides (Thermo Scientific Inc.) and dried for at least 24 h at room temperature.

The sections were dewaxed, dehydrated and stained automatically with haematoxylinand eosin (H&E) (Varistan 24-2, Shandon GmbH, Frankfurt, Germany) in accordance with a protocol (Table 3). Subsequently, the sections were covered with ROTI^®^Mount and coverglass, dried for at least 24 h and examined microscopically.

For the histological specimens, a scoring key of 1–5 was used for semiquantitative analysis of the inflammatory infiltrate for microscopic examination. Specimens were graded according to the degree of neutrophilic and increased lymphoplasmacytic cell infiltration, and possible differences between groups were evaluated (1 = slight infiltration, 2 = mild infiltration, 3 = moderate infiltration, 4 = marked infiltration, 5 = severe infiltration).

### 2.6. Dry Matter Determination of Ingesta (Caecum)

To determine the dry matter (DM) of ingesta from the caecum, approximately 1 g of the sample (E) was weighed into a metal crucible (T1) and then dried at 103 °C for a period of at least four hours in a drying oven.

The samples were allowed to cool in a desiccator before the porcelain crucibles were weighed again (T2). Ten samples were treated per group.

The following formula was used to determine the dry matter content.
$$\mathrm{Calculating}\;\mathrm{of}\;\mathrm{DM}\;(g/\mathrm{kg})=\left[\frac{T1-T2}{E}\right]\times 1000$$

### 2.7. Data Analysis and Statistics

Results from the Ussing chamber experiment were expressed as the mean value ± standard deviation (MW ± SD) of the single values of the three chambers used per bowel segment. For the calculation of ∆I_sc_, the maximum changes after adding the respective substance were used. The statistical analysis for the results of the Ussing chamber, histological samples and dry matter determination was performed with a one-way ANOVA and the post-hoc Dunnetts *t*-test using SAS v. 9.1 (SAS Inst. Inc., Cary, NC, USA).

The differences were assumed to be statistically significant if *p* ≤ 0.05.

## 3. Results

### 3.1. Clinical and Microbiological Findings during the Test Period and Macroscopic Findings in the Section

No abnormalities in the general condition of the animals were observed during the experimental period. Diarrhoea or increased internal body temperature did not occur (Appendix A). Faecal samples of the control group (Group 0) remained *Campylobacter* negative throughout the trial. In the infected groups, the respective strains could be detected in the faeces as early as two days after infection (p. inf.) in some cases, but reliably seven days after inoculation. In Group 2, the detection rate decreased slightly again after day 7 (Table 4). Cross infection with the other *Campylobacter* strain was not detected in Groups 1 and 2.

On dissection, there were no macroscopic changes to the intestinal tract, liver, or spleen.

### 3.2. Distribution of Campylobacter spp. in Different Intestinal Segments

Cultural investigation of the gut demonstrated different detection frequencies of the *Campylobacter* strains in the intestinal sections (Table 5).

*C. coli* as well as *C. jejuni* could be reisolated from all samples from the caecum and colon. This also applied for mono- and co-infection (Groups 1–3).

Only *C. jejuni* was found in the duodenum. In Group 2, 40% (*n* = 4) of the samples were positive, while in Group 3, 62.5% (*n* = 5) were positive. *C. coli* was not detected in the duodenum in either mono- or co-infection.

*C. jejuni* could be isolated from the jejunum samples for Group 3 in all pigs. However, only 12.5% of the samples showed *C. coli* (*n* = 1). Moreover, in the mono-infected pigs, the detection of *C. jejuni* in the jejunum was higher (Group 2; 70%; *n* = 7) than that of *C. coli* (Group 1; 25%; *n* = 2).

*C. jejuni* was detected in the ileum of all animals in Groups 2 and 3. In contrast, the identification of *C. coli* showed a difference between mono- and co-infection. While *C. coli* was detected in the ileum of all pigs in Group 1, only 12.5% of the animals in Group 3 were positive (*n* = 1).

For Group 0, the *Campylobacter* negative status was confirmed by the negative results of the microbiological investigation of all intestinal sections.

### 3.3. Basal, Glucose-Induced and Forskolin-Induced Short-Circuit Currents (I_sc_) in Jejunal Epithelia

After equilibration, the basal values for I_sc_ were at a similar level in all four groups (Figure 1a). An increase in I_sc_ could be measured after adding glucose and forskolin in all groups (Figure 1a) without a statistically significant difference between them (*p* > 0.05). The course of short-circuit currents and tissue conductance during the experimental period in the jejunum in an animal of the control group (Group 0) is exemplarily shown in Figure 2. The time course was similar for animals in the infected groups (all values can be found in Appendix A).

### 3.4. Basal, Carbachol-Induced and Forskolin-Induced Short-Circuit Currents (I_sc_) in Caecal Epithelia

Between Group 1, Group 3, and the control group (Group 0), no statistical differences in the I_sc_ of the caecal epithelia could be demonstrated. The basal caecal I_sc_ in Group 2, mono-infected with *C. jejuni*, was statistically significantly higher (*p* < 0.05) than in Group 0. After adding glucose and forskolin, the I_sc_ in Group 2 continued to differ statistically significantly from Group 0 (Figure 1b and Figure 3; detailed data in Appendix A).

### 3.5. Conductance (G_t_) of Jejunal and Caecal Epithelia

The basal G_t_ values in jejunal and caecal samples did not differ statistically significantly between groups. The changes after adding glucose to the jejunal tissue and carbachol to the caecal tissue were also not statistically significantly different (*p* > 0.05). Accordingly, adding forskolin 30 min later did not result in statistically significant changes in the G_t_ values (*p* > 0.05). A time course of the changes in the G_t_-value in caecal tissue is shown in Figure 3 (detailed data in Appendix A).

### 3.6. Results of the Histological Examination

Almost all tissue samples of the intestinal sections (duodenum, jejunum, ileum, caecum, and colon) of all pigs were given the scores 1 or 2 in the applied scoring system, corresponding to a slight to mild neutrophilic and increased lymphoplasmacytic cell infiltration. Only two samples from the duodenum (both from Group 2) were found to have a score of 3. No statistically significant differences between the infected groups (Groups 1, 2, and 3) and the control group (Group 0) were detected (*p* > 0.05).

### 3.7. Dry Matter of the Caecal Content

Due to the differences between Group 2 and Group 0 in the tests using the Ussing chamber, a dry substance determination from the caecal content was carried out. Samples from each group were averaged and compared (Group 0: 143.52 ± 17.68; Group 1: 135.86 ± 37.28; Group 2: 108.87 ± 17.23; Group 3: 139.29 ± 28.33). The dry matter content of the caecum from Group 2 was statistically significantly lower than that from the control group (*p* < 0.05).

## 4. Discussion

The aim of this study was to evaluate whether colonisation of weaning pigs with different *Campylobacter* strains has an impact on intestinal health and mucosal barrier function.

Prerequisite for the *Campylobacter* infection model was the rearing of the piglets without infection with *Campylobacter* field strains, which was confirmed by the absence of *Campylobacter* spp. before inoculation and in the control group. In experiments with gnotobiotically reared piglets [9,10,11], a method for maintaining absence from *Campylobacter*-strains for days up to a few weeks was established in a highly sterile environment. In the present study, it was shown that *Campylobacter*-free rearing over a period of 12 weeks after Cesarean section is possible under strict hygiene standards without sterile conditions.

In previous studies, it was observed that a low infectious dose of 800 cfu was sufficient to cause disease in 50% of the subjects in humans. At a dose of 10^8^, the disease rate was 100% [29,30].

An infectious dose of at least 10^8^ cfu per animal was obtained. In previous studies, this infectious dose has been shown to be sufficient to achieve intestinal colonisation in pigs as safely as possible [28,31].

Although the piglets were assumed to have no maternal immunity against *Campylobacter*, comparable to colostrum-deprived or gnotobiotic piglets, none of them developed clinical signs throughout the experiment, but were successfully colonised. Furthermore, during post-mortem examination, internal organs did not show any gross lesions.

These observations are consistent with those of previous studies [28,31,32]. For the most part, *Campylobacter* spp. are considered commensals of the swine intestine [4]. Thus, as previously mentioned, it could be demonstrated, that *Campylobacter* spp. have no clinical effects on pigs. Therefore, the importance of *Campylobacter* infections in diarrhoel diseases of pigs can be neglected. However, when colonising gnotobiotic piglets with *C. jejuni* on the day of birth, significant clinical abnormalities were observed [9], comparable to other *Campylobacter* infection models with very young pigs [10,11]. In contrast to the histopathological results of the presented study, other trials with gnotobiotic piglets showed distinct histological alterations of the intestinal epithelium. These changes were described as an increase in mucous secretory cells, signs of villi shortening and blunting. In addition, some villous cells showed damaged nuclei and some crypt abscesses were present in the caecum [9,10]. These changes were not found in the present study. Thus, it seems that the intestine of gnotobiologically reared animals is more affected by *Campylobacter* spp. colonisation than is the case in animals that have not been kept under sterile conditions.

The Ussing chamber technique allows the measurement of the electrophysiological tissue parameters (conductance (G_t_)) and short circuit current (I_sc_). In order to eliminate electrochemical gradients across the tissues, the experiments were performed under short-circuited conditions, so that a change in short circuit current (I_sc_) implies a change in net transepithelial ion transfer. Additionally, tissue conductance (G_t_) provides an indication of epithelial integrity [33].

Changes in short circuit currents depend on the transepithelial ion transport. Adding glucose to the mucosal side of jejunal samples stimulates the electrogenic SGLT1 transport system, resulting in an increase in I_sc_ [22,23].

For caecal tissues, the addition of glucose was replaced by carbachol, since glucose is almost exclusively absorbed in the small intestine [23,34]. Forskolin and carbachol stimulate Cl^−^ secretion: forskolin activates cAMP-controlled secretion [24] whereas carbachol acts on Ca-controlled secretion [25,26,27]. This ion release into the intestinal lumen is always passively followed by H_2_O. Thus, when the secretion of Cl^−^ is increased, a greater amount of water is also released into the intestinal lumen. This manifests itself clinically as diarrhea [35].

It can be concluded that jejunal glucose transport by the secondary active co-transporter 1 (SGLT1 transport) [36,37] was not affected by *Campylobacter* colonisation. Similarly, cAMP-mediated chloride secretion was not altered. The I_sc_ increase from the jejunal epithelia was not statistically significantly different among all groups after adding forskolin (*p* > 0.05).

Microbiological investigation of ingesta showed that predominantly *C. jejuni* seems to colonise the jejunum, although *C. coli* was also detected, but to a lesser extent, which has also been demonstrated in a previous study [28]. Both agents did not affect the integrity of the studied intestinal barrier and the transport mechanisms.

Investigations on the caecal tissue showed that only Group 2 (mono-infected with *C. jejuni*) was found to differ statistically significantly from the control group in forskolin- and carbachol-induced increases in I_sc_ (*p* < 0.05) due to an increased permeability for ions (Cl^−^). Forskolin induces cAMP-regulated chloride secretion, whereas carbachol acts on calcium-regulated chloride secretion. In both mechanisms, water passively follows chloride into the intestinal lumen [27], initially causing a more fluid ingesta. Dry matter determination of caecal content confirmed these findings, although clinically, diarrohea could not be detected. One potential explanation for this fact might be that a substantial amount of water is reabsorbed along the intact colon [34], which could compensate the water loss in the caecum. The fact that the colon is an intestinal segment with a high water resorption rate has already been extensively described [34]. The mechanism of the caecal epithelium alteration remains unclear.

In connection with this, it can be speculated, that an infection with *C. jejuni* could have a subclinical effect on pigs and that the loss of water in the caecum could also have an influence on nutrient intake. This can lead to losses in performance and weight development, which could well be decisive in conventional pig farming. To be more precise, longer-term studies should be carried out to study the weight development of the pigs in more detail. In earlier experiments by Leonhard-Marek et al. [38], pigs were infected with parasites. It was found that chloride secretion was increased during the penetration phase of the parasites into the intestinal mucosa. It is possible that *C. jejuni* has a similar effect on the intestinal barrier, resulting in an increase in chloride secretion.

Despite the differences observed in the Ussing chamber experiments, corresponding histological changes could not be found, leading to the assumption that colonisation of *C. jejuni* affects the transport mechanisms of the intestinal mucosa without any inflammatory reaction. Van Deun et al. (2008) demonstrated in an in-vitro experiment that *C. jejuni* strains could invade chicken primary caecal epithelial crypt cells without showing any signs of necrosis or intestinal inflammation [39]. Transmigration might be possible by the cleavage of occludin, a tight junction protein impairing intestinal barrier function, which is assumed by Harrer et al. [40]. Further studies are needed to investigate the mechanism of *Campylobacter* spp. concerning disturbance of the intestinal barrier in swine. In addition, further studies should clarify whether the subclinical effect that the *C. jejuni* strain used in the present study has on pigs can be transferred to other *C. jejuni* strains or only applies to the one used here.

No significant increase in I_sc_ was found either in mono-infected pigs with *C. coli* or in co-infected animals with *C. coli* and *C. jejuni*. A previous study by Leblanc et al. [32] proved that *C. coli* has a higher colonisation potential than *C. jejuni* in pigs. This could result in *C. coli* overriding *C. jejuni* colonisation in co-infection to the extent that a smaller amount of *C. jejuni* colonising the caecum did not have any impact on the intestinal transepithelial chloride transport. Enumeration of *Campylobacter* spp. in the ingesta may be able to substantiate this assumption.

## 5. Conclusions

The results of the presented study demonstrate that the inoculated *C. coli* and *C. jejuni* strains of poultry origin are capable of colonising the intestine of weaned pigs and that both species integrated well after co-infection. The pigs did not show any clinical signs of disease after inoculation, although investigations using the Ussing chamber demonstrated an impact of *C. jejuni* on the caecal epithelium and the chloride transport mechanism. However, this effect appears to be compensated by the intact colon of the piglets so that no clinical disease occurs. Additionally, this study showed that the raising of *Campylobacter* free pigs is possible under high hygienic standards and that this pig colonisation trial seems to be suitable for studying *Campylobacter* infections without the need for gnotobiotic or specific-pathogen-free (SPF) animals.

## Figures and Tables

**Figure 1 animals-11-02742-f001:**
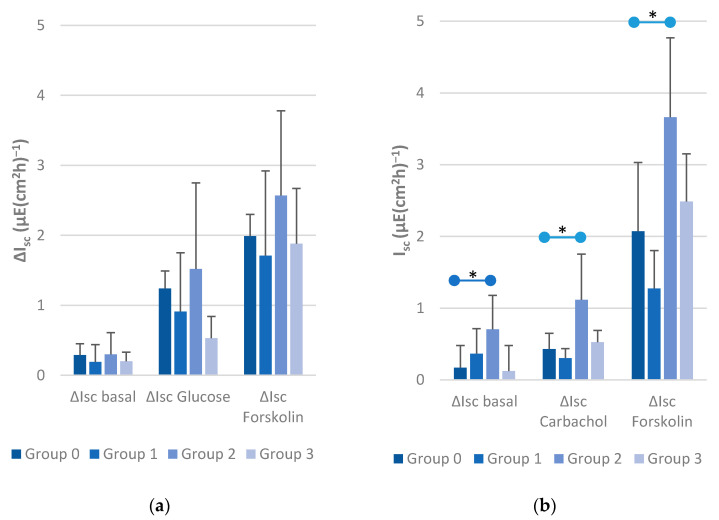
Basal short-circuit currents and I_sc_ changes of jejunal and caecal epithelia after adding glucose and forskolin. (**a**) Measurements of the Ussing chamber experiment with jejunal epithelia. Indicated are the basal short-circuit currents and the changes after adding 10 mmol glucose (100 µL, mucosal) and 5 × 10^−6^ M forskolin (1 µL, serosal). (**b**) Measurements of the Ussing chamber experiment with caecal epithelia. Indicated are the basal short-circuit currents and the changes after adding 10^5^ M carbachol (10 µL, serosal) and 5 × 10^−6^ M forskolin (1 µL, serosal); statistically significant values are marked with * (*p* < 0.05).

**Figure 2 animals-11-02742-f002:**
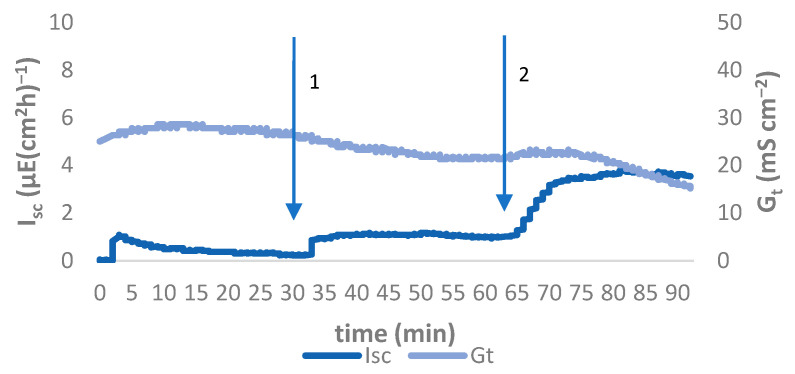
Typical time course of I_sc_ and G_t_ of jejunal epithelium using Ussing chamber experiment of a pig in Group 0. Arrows indicate addition of pharmaceuticals: 1 glucose (mucosal) + mannitol (serosal), 2 forskolin (serosal).

**Figure 3 animals-11-02742-f003:**
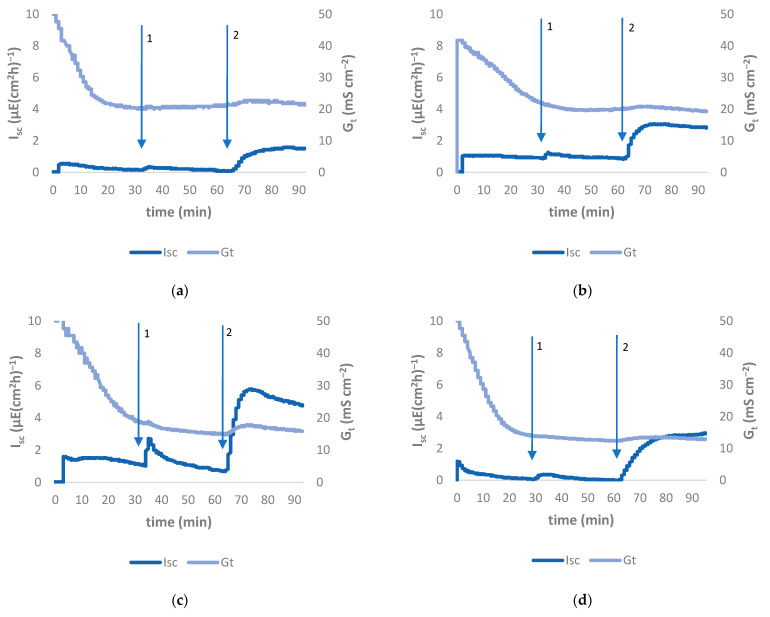
Time courses of I_sc_ and G_t_ in caecal tissue using Ussing chamber experiment of (**a**) Group 0, (**b**) Group 1, (**c**) Group 2, (**d**) Group 3. Arrows indicate addition of pharmaceuticals: 1 carbachol, 2 forskolin. The time course of the I_sc_ of Group 2 (**c**) differs statistically significantly from Group 0 (*p* < 0.05).

**Table 1 animals-11-02742-t001:** Infection scheme of the animal groups.

Group	Number of Piglets	Inoculation	Inoculation Dose/10 mL (cfu)
0	11	10 mL nutrient broth	0
1	8	10 mL nutrient broth with *C. coli*	1 × 10^8^
2	10	10 mL nutrient broth with *C. jejuni*	5 × 10^9^
3	8	10 mL nutrient broth with *C. coli* +10 mL nutrient broth with *C. jejuni*	1.9 × 10^10^ *C. Coli* 7.7 × 10^8^ *C. jejuni*

**Table 2 animals-11-02742-t002:** Protocol for the preparation of buffer solutions for the Ussing chamber procedure (concentration in mmol/L).

Chemicals	Serosal Side [mmol/L] (Both Epithelia)	Mucosal Side [mmol/L] (Jejunal Epithelia)	Mucosal Side [mmol/L] (Caecal Epithelia)
NaCl	113.6	113.6	53.6
KCl	5.4	5.4	5.4
HCl	0.2	0.2	0.2
MgCl_2_	1.2	1.2	1.2
CaCl_2_	1.2	1.2	1.2
NaHCO_3_	21.0	21.0	21.0
Na_2_HPO_4_	1.5	1.5	1.5
glucose	10.0	-	-
mannitol	2.0	2.0	2.0
HEPES ^1^	7.0	20.0	10.0
Na-gluconate	6.0	-	6.0
NaOH	-	6.0	-
Na-acetate	-	-	36.0
Na-propionate	-	-	15.0
Na-butyrate	-	-	9.0

^1^ (4-(2-hydroxyethyl)-piperazine-1-ethanesulfonic acid).

**Table 3 animals-11-02742-t003:** Protocol for staining the histological samples.

Process	Chemicals	Time
Dewaxing and rehydration	N-butyl acetate	2 × 10 min
	Isopropanol 100%	5 min
	Isopropanol 96%	5 min
	Isopropanol 70%	5 min
	Isopropanol 50%	5 min
Hemalaun staining	Hemalaun Usage Solution	10 min
Washing	Running tap water	10 min
	Distilled water	2 min
Eosin staining	Eosin working solution	5 min
Washing	Distilled water	2 × 30 s
Dehydration	Isopropanol 70%	30 s
	Isopropanol 70%	2 min
	Isopropanol 70%	2 min
	N-butyl acetate	10 min

**Table 4 animals-11-02742-t004:** Percentage of positive samples in weekly faecal sampling for detection of the inoculated *Campylobacter* strain before dissection over a four-week period (Group 1: *n* = 8; Group 2: *n* = 10; Group 3: *n* = 8).

Days p. inf.	Group 1 (%)	Group 2 (%)	Group 3 (%)	
			*C. coli*	*C. jejuni*
2	87.5	30	25	37.5
7	100	100	100	100
14	100	90	100	75
21	100	70	100	100

**Table 5 animals-11-02742-t005:** Isolation frequency of *Campylobacter* spp. in different intestinal sections after dissection in the twelfth week of life. *C. coli* mono-infection in Group 1; *C. jejuni* mono-infection in Group 2; co-infection divided into detection of *C. jejuni* and *C. coli* (Group 1: *n* = 8; Group 2: *n* = 10; Group 3: *n* = 8).

Intestinal Section	Group 1 (%)	Group 2 (%)	Group 3 (%)	
			*C. coli*	*C. jejuni*
Duodenum	0	40	0	62.5
Jejunum	25	70	12.5	100
Ileum	100	100	12.5	100
Caecum	100	100	100	100
Colon	100	100	100	100

## Data Availability

The data presented in this study are available in Appendix A. Additional data are available on request from the corresponding author.

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
