# Peer review of "Impact of Campylobacter spp. on the Integrity of the Porcine Gut"

_animals, 2021, doi:10.3390/ani11092742_

Round 1
Reviewer 1 Report
Animals Manuscript ID: 1336452
Independent Review report
Brief Summary: This original research paper describes how Campylobacter colonization in the pig gut is affected by transport mechanisms in the intestine and cecum. The paper’s novelty was the authors’ approach to inoculate pigs with one or two species of Campylobacter to observe their colonization effects on the gut electrophysiology. The paper’s strength is the data reported provides evidence that Campylobacter’s colonization is strongly related to the host’s intestinal lining transport system and elicits further questioning with why inflammatory elements of leukocyte infiltration was not affected. There are a few areas of improvement that can increase this paper’s quality and impact for readers. The Introduction section starts off strong, but is lacking depth with providing readers background terminology necessary for understanding the following sections. The Discussion section can be improved by including justifications on why the data provided by the paper can drive the field with either future applications or advancements in the clinical side. Lastly, although it was stated in the acknowledgments that the paper was proofread by an English editor, the paper has a number of grammatical mistakes that should be addressed. This exciting paper has valuable insights for porcine research and its value as a clinical model for advancing gastrointestinal health research!!
Major comments:
This paper has wonderful tables that are easy to follow!
Introduction: The section needs depth with setting up readers with important key terms for understanding the paper. Forskolin and carbachol are mentioned in the Materials and Methods and Results, but the terms are not described until Lines 387 – 389. Is it possible to move that sentence to the Introduction? The sentence is outstanding and gives readers context with what to expect when they see the words later in the paper.
Tables and Figures: For tables and figures that provide results, it would help if the authors would consider adding a brief note describing the sample size used to generate that data for that table.
Also, acronyms used in figures and tables should be defined in each corresponding caption because each figure and table should be able to stand on its own without support from the main text and some readers may look at individual figures or tables separate from the text. This comment also applies to the Supplementary Materials as well.
Discussion: This section has good details, but could be more impactful with including a few sentences that highlight how the paper’s findings can advance the field, especially from the clinical standpoint.
Minor Comments:
Line 118: “eighth” should be changed to “eight” or phrased as “in the eighth week” like in Line 122.
Line 132: What is “per os” supposed to be?
Line 222: Is “haematoxilyn” supposed to be “haematoxylin?
Table 4: It is understandable why the 'n=x' was used to show the absolute values. However, it is confusing because it can easily be assumed to be the sample size. Is it possible for the authors to omit that information and instead, include the n=x in the table caption for the sample size for each group?
Figure 1: Is it possible to convert this figure to a table? I believe the table would provide the same information as the figure, but the benefit is delivering the information on a simpler scale to be easier for readers to interpret. It can look like Table 4 with the first column used for tissue sections with each row in that column as the corresponding tissue section. Similar to my major comment regarding tables and figures, providing a statement on the sample size would be helpful for readers, but also including a note on the day of age of the pigs when the samples were collected.
Figure 2: Is it possible to add a bar, with the asterisk on top of it, above the corresponding groups to denote they are statistically different? While the asterisk is currently set to show that, readers can potentially interpret it differently.
Otherwise, the figure and the description are outstanding!
Line 371 – 373: Why were the alterations noted in these lines not mentioned in the results section? Also, these statements are confusing. Were all these changes observed in this study? The citations are fine, but these sentences should be revised to make it clearer on what was observed in the study versus what was observed in other studies.
Line 377 – 381: This is a fantastic paragraph that describes why the Ussing chamber was used in this study!
Lines 444 - 445: It is not clear on how the point in this sentence relates to the overall objectives pursued by the paper. The authors should provide supporting details in the Discussion to highlight this point.
Supplementary Materials
Table 1, 2, 3: What is “Aqua bidest” supposed to be? And for Tables 2 and 3, for the second column, what is “ad” supposed to be?
Reviewer 2 Report
Dear authors,
The present work "Impact of Campylobacter spp. on the integrity of the porcine gut" is well elaborated with a well described methodology and results are supported by literature. In my view this study complements previous work on pathogenicity mechanisms of Campylobacter species.
My recommendation is to accept with minor changes.
Minor changes:
- page 2 line 57: intestinal flora - although a common mistake, intestine is colonized by microbes and not plants. Please change to: intestinal microbiota;
- page 3 line 111 and page 5 line 219: the copyright symbol should be in superscript;
- page 3 line 124: missing G in GmbH;
- page 3 line 141: 13th - the "th" should be superscript;
- page 4 line 164 and 165: O2, CO2 and N2 - numbers should subscript;
- page 5 line 196: mmol/l should be mmol/L;
Reviewer 3 Report
This is a very nice, well written article detailing the effect which Campylobacter infection has on the gut on pigs. The experiments are well performed and explained, and I only have a few very minor comments and questions below.
Line 20- you don’t need (C)
Line 28- a comma after pigs may improve flow
Line 41- a comma after therefore may improve flow
Line 48-49- it may be worth saying that this is in humans
Line 79- would these results differ if a C. coli strain which was adapted to pigs was used instead of a chicken one?
Line 94- should this be spp.?
Line 114- ad libitum should be in italics
Line 197- comma after gradients
Line 198- comma after mins
Line 248- maybe write in the programme which was used for stats analysis
Line 259- were cross infection tests done- i.e. did you see if the C. coli group had C. jejuni etc? (You may have said it but I wasn’t sure)
Line 270- given that not all animals were infected, were these uninfected tissues removed from the analysis? It may have skewed your results is my only thinking
Line 379- comma after tissues
Line 403- perhaps word to …’was found to have a statistically significant difference…’?
Line 420 -in vitro should be in italics
As mentioned above, these comments are only minor and shouldn’t detract from this being a very well written, well performed and interesting article.
I wish to commend the authors, and wish them all the best for their future research.
Round 2
Reviewer 1 Report
Outstanding job with the revision!
I have two minor comments that should be easy to address for type editing:
Line 273: What does "thereof were positive" supposed to mean?
Supplementary Table 6: In the first column, to keep it consistent with the other tables, group should be capitalized, except for the group used in the column heading.
Once again, awesome job with this exciting work!